# *MindZero*: Learning Online Mental Reasoning With Zero Annotations

**Shunchi Zhang** [1] [*]  **Jin Lu** [1] [*]  **Chuanyang Jin** [1] [*]  **Yichao Zhou** [2] [*]  **Zhining Zhang** [2]  **Tianmin Shu** [1]

https://scai.cs.jhu.edu/MindZero
 Code   Data   Model

## Abstract

Effective real-world assistance requires AI agents with robust Theory of Mind (ToM): inferring human mental states from their behavior. Despite recent advances, several key challenges remain, including (1) online inference with robust uncertainty updates over multiple hypotheses; (2) efficient reasoning suitable for real-time assistance; and (3) the lack of ground-truth mental state annotations in real-world domains. We address these challenges by introducing *MindZero*, a self-supervised reinforcement learning framework that trains multimodal large language models (MLLMs) for efficient and robust online mental reasoning. During training, the model is rewarded for generating mental state hypotheses that maximize the likelihood of observed actions estimated by a planner, similar to model-based ToM reasoning. This method thus eliminates the need for explicit mental state annotations. After training, *MindZero* internalizes model-based reasoning into fast single-pass inference. We evaluate *MindZero* against baselines across challenging mental reasoning and AI assistance tasks in gridworld and household domains. We found that LLMs alone are insufficient; model-based methods improve accuracy but are slow, costly, and limited by backbone MLLM capacity. In contrast, *MindZero* enhances MLLMs' intrinsic ToM ability and significantly outperforms model-based methods in both accuracy and efficiency, showing that mental reasoning can be effectively learned as a self-supervised skill.

---
[*]Equal contribution  [1]Johns Hopkins University [2]Peking University. Correspondence to: Tianmin Shu <tianmin.shu@jhu.edu>.

*Proceedings of the 43rd International Conference on Machine Learning*, Seoul, South Korea. PMLR 306, 2026. Copyright 2026 by the author(s).

## 1. Introduction

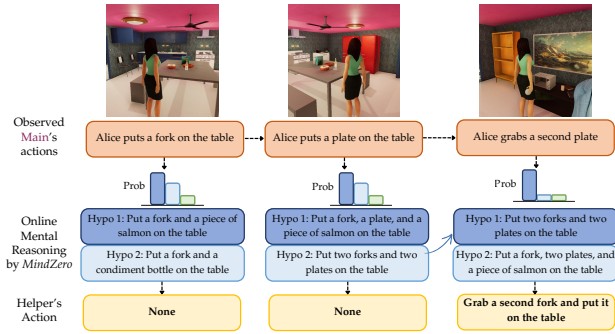

*Figure 1.* An example of online mental reasoning for proactive assistance, where the helper agent simultaneously infers the the main agent's goal and helps to reach the goal faster. As shown in this example, the helper observes the main agent's actions over time, *MindZero* continuously updates a probability distribution over multiple goal hypotheses. Based on the multiple possible hypotheses maintained at each step, the helper decides whether to act and proactively assists by fetching relevant tableware and placing it on the table. As new actions are observed, the probabilities of different mental state hypotheses are updated over time. In particular, the transition from step 2 to step 3 shows that the main agent grabbing a second plate increases the likelihood of the second hypothesis at step 2.

To proactively assist human users in the real world, AI agents must understand users' minds and anticipate their needs. This requires strong Theory of Mind (ToM), i.e., the ability to infer users' mental states (such as desires, beliefs, and goals) from their behavior. Recent advances in large language models (LLMs) and multimodal LLMs have sparked growing interest in machine Theory of Mind (Wimmer & Perner, 1983; Ullman, 2023; Wilf et al., 2024; Sclar et al., 2023; Jin et al., 2024). However, much of the existing work focuses on question-answering-based ToM evaluation and development, which is insufficient for real-world assistance. In practice, an assistive agent must continuously update its inferences about a user's mental state and track uncertainty over multiple competing hypotheses. This form of online mental-state reasoning can guide agent planning, enabling proactive assistance, adaptation to changing contexts, and more effective collaboration with users.

For instance, in Figure 1, as the agent observes a human's actions in a household setting, it maintains and updates a probability distribution over multiple possible goal hypotheses in real time, and uses these hypotheses to decide when and how to proactively help (e.g., fetching tableware before the user asks).

However, training models for online mental reasoning remains challenging. Human mental states are latent and often ambiguous. They are also dynamically changing over time in sequential tasks. For many real-world applications, such as household or web assistance, it is extremely difficult and costly to collect large-scale training data with reliable annotations of ground-truth mental states. As a result, prior works on learning-based ToM methods have been limited to controlled settings (Rabinowitz et al., 2018; Rhinehart et al., 2019; Bortoletto et al., 2024a;b), lacking open-endedness and scalability.

To circumvent these data and annotation challenges, recent work has explored inference-time reasoning methods that leverage the generality and strong reasoning ability of LLMs for ToM, without requiring model training. In particular, when integrated with model-based ToM methods, such as Bayesian inverse planning (BIP), inference-time scaling has demonstrated strong performance on challenging ToM reasoning tasks (Jin et al., 2024; Shi et al., 2025; Zhang et al., 2025; Ying et al., 2023; Kim et al., 2025). These methods leverage LLMs to propose and evaluate mental state hypotheses, achieving robust and scalable mental reasoning. However, they are computationally prohibitive in online mental reasoning required for real-world assistance tasks. These challenges call for a new type of ToM approach that retains the deliberative structure of model-based reasoning while better leveraging the efficiency and learning capacity of LLMs.

To address these limitations, we introduce *MindZero*, a novel Theory of Mind reasoning framework that trains multimodal language models to perform robust and efficient online mental reasoning without requiring mental state annotations. During training, the model explicitly generates hypotheses about mental states (e.g., beliefs and goals) and is rewarded when these hypotheses assign high likelihood to the actions people actually take. We term this Self-Supervised Reinforcement Learning (SSRL). Unlike common RL-based language model training, the reward in our SSRL method is computed entirely from self-supervised signals. It encourages the model to produce explicit mental state hypotheses with robust uncertainty estimates. This method eliminates the need for ground-truth mental state labels, allowing the model to learn directly from behavior and internalize ToM reasoning patterns that explain actions in context. The trained *MindZero* model infers mental states in a single forward pass, while remaining grounded in a model-based objective that preserves robustness and interpretability.

In our experiments, we compared *MindZero* against state-of-the-art ToM methods on question answering and proactive assistance tasks in both gridworld (Jha et al., 2024) and household environments (Puig et al., 2023). Small multimodal language models trained with our *MindZero* method significantly outperformed baselines in all tasks, matching the robustness of model-based methods while significantly reducing the computational cost. We further validate *MindZero* in an IRB-approved human study, where it delivers effective real-time assistance to human users using a small open-weight backbone. These results suggest that mental reasoning can be learned as a self-supervised skill, narrowing the gap between robust but slow model-based inference and fast but error-prone reasoning by a small multimodal language model.

In sum, our main contributions include: (1) a self-supervised RL method, *MindZero*, that trains multimodal language models to conduct robust and efficient online mental reasoning without mental state annotations; (2) systematic evaluation of *MindZero* and recent ToM methods in a suite of challenging online mental reasoning and proactive AI assistance benchmarks.

## 2. Related Work

**Theory of Mind Methods.** Existing methods for ToM reasoning fall into three main categories. (1) *Prompting-based* approaches (Jung et al., 2024; Huang et al., 2024; Yu et al., 2024; Zhou et al., 2025a; Hou et al., 2024; Sclar et al., 2023) improve upon base LLMs but still exhibit systematic errors in long-context understanding, complex behaviors, and recursive reasoning. (2) *Model-based* approaches, especially Bayesian inverse planning (BIP) (Baker et al., 2009; Ullman et al., 2009), explicitly model agents' mental states and their influence on behavior. Recent work integrates BIP with LLMs (Jin et al., 2024; Shi et al., 2025; Zhang et al., 2025), combining structured reasoning with flexible language understanding. However, these methods are often computationally expensive, as they require searching large hypothesis spaces at test time. (3) *Learning-based* methods train neural networks for mental-state inference (Rabinowitz et al., 2018; Liang et al., 2024; Sclar et al., 2024; Lu et al., 2025), but they rely on costly and unreliable ground-truth annotations, limiting their scalability and applicability. To address these limitations, *MindZero* learns mental reasoning directly from human behavior data. Our approach improves over prompting-based methods, avoids the computational overhead of model-based inference, and eliminates the need for explicit mental state annotations required by prior learning-based approaches.

**ToM-Guided Assistance**    Recent work on ToM has been mainly focused on question-answering tasks (Le et al., 2019; Gandhi et al., 2023; Kim et al., 2023; Wu et al., 2023; Xu et al., 2024; Jin et al., 2024; Shi et al., 2025; Bortoletto et al., 2025a; Fan et al., 2025), where ToM models answer questions about mental states based on a story and/or a video. In contrast, ToM-guided assistance is more challenging: models must continuously infer and update mental states while accounting for uncertainty over long horizons to support effective assistance. Prior work has explored Theory of Mind guided assistance (Puig et al., 2023; Ying et al., 2024; Zhi-Xuan et al., 2024; Zhou et al., 2025b; Jin et al., 2025; 2026) where an agent helps a human based on its understanding of the human's mind across domains such as games, household environments, coding, and real-world LLM conversations. Other work studies assistants supporting teams with shared goals (Seo et al., 2023; Zhang et al., 2024) or partially divergent goals (Bortoletto et al., 2025b) through intervention and coordination. A further line focuses on situated natural-language collaboration with rich social dynamics (Liu et al., 2012; Chai et al., 2014; Suhr et al., 2019; Narayan-Chen et al., 2019; Jayannavar et al., 2020; Bara et al., 2021; Bortoletto et al., 2025a). Although there has been prior work on online mental reasoning shown to be effective in ToM-guided assistance (e.g., Puig et al., 2023; Wang et al., 2021; Shvo et al., 2022; Zhi-Xuan et al., 2024; Ying et al., 2024; Cross et al., 2024; Ma et al., 2025), they have strong assumptions about human behavior and/or require high computational costs for complex tasks. *MindZero* directly targets this gap by training a small multimodal language model to efficiently and robustly conduct online mental reasoning that can support downstream assistance tasks in a scalable way.

# 3. Problem Formulation

We formalize the problem of online mental state inference (Section 3.1) and characterize how inferred mental states can be leveraged to enable proactive assistance (Section 3.2). Our formulation provides a unified probabilistic framework for reasoning about users' latent beliefs and goals from sequential observations, and for translating this uncertainty-aware reasoning into effective assistive decision making in dynamic environments.

## 3.1. Online Mental Reasoning

Given a sequence of observed user behavior up to time step $t$, including states $s_{1:t}$ and actions $a_{1:t}$, a ToM model infers the latest mental state of the user $m_t$, which could include different mental variables such as beliefs $b_t$ and goals $g_t$. Inspired by Bayesian inverse planning (BIP) (Baker et al., 2009; 2017; Zhi-Xuan et al., 2020), a model-based ToM inference method, we formalize online mental state inference

as following Bayesian inference:

$$\underbrace{P(m_t \mid s_{1:t}, a_{1:t})}_{\text{posterior}} \propto \underbrace{P(a_{1:t} \mid m_t, s_{1:t})}_{\text{action likelihood}} \cdot \underbrace{P(m_t)}_{\text{prior}}, \quad (1)$$

Unlike prior work by (Zhi-Xuan et al., 2020), this formulation goes beyond the typical Markovian assumptions behind BIP, modeling all past behavior jointly. In real-world domains, this Bayesian inference can be computationally intractable due to an infinite hypothesis space and costly action likelihood estimation (which is achieved via forward planning conditioned on hypothetical mental states). Our *MindZero* method aims to overcome these computational bottlenecks by training a multimodal language model to directly output quality hypothesis samples and their posterior probabilities without explicit Bayesian inference.

## 3.2. Proactive Assistance Guided by Online Mental Reasoning

In online mental reasoning, the model must continuously update multiple mental state hypotheses $\{m_t\}$ at every step $t$ and estimate their probabilities $\{q_t\}$ given a user's behavior history $(s_{1:t}, a_{1:t})$. Given the top hypotheses of a user's mental state, an assistive agent can then plan for the assistive actions to best help the user. Let $a_t^A$ be the assistive action at time step $t$. We define the assistive agent's policy as

$$P(a_t^A \mid s_{1:t}, a_{1:t}) = \sum_{m_t} P(a_t^A \mid s_t, m_t) P(m_t | s_{1:t}, a_{1:t}). \tag{2}$$

Such assistive decision making must consider the uncertainty in the mental inference, which requires a robust estimate of the confidence of multiple hypotheses. It also needs to frequently update plans based on the most recent user behavior, and thus needs a fast inference to support real-time replanning. *MindZero* aims to achieve this via training a small multimodal language model with low computational cost and latency.

# 4. *MindZero*

We introduce *MindZero*, a self-supervised reinforcement learning framework that trains multimodal language models to perform efficient and robust online mental reasoning. *MindZero* learns directly from behavioral data using self-supervised signals, addressing the lack of ground-truth mental state labels in real-world domains (Section 4.1 and Figure 2a). The core of *MindZero* is its reward design: the model is rewarded for generating mental state hypotheses that maximize the likelihood of observed actions, as estimated by a model-based planner or an LLM, in a manner similar to model-based ToM reasoning (Section 4.2 and

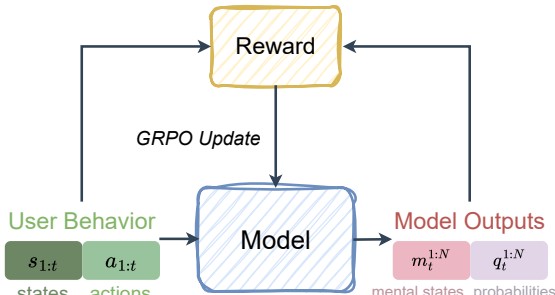
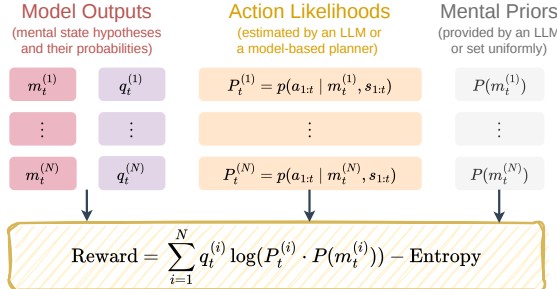

*(a)* Self-Supervised Reinforcement Learning.  *(b)* Reward Computation.

*Figure 2.* (a) Overview of our Self-Supervised Reinforcement Learning (SSRL) framework. Given **states** $s_{1:t}$ and **actions** $a_{1:t}$ up to timestep $t$, the model outputs a set of $N$ **mental state hypotheses** $m_t^{1:N}$ along with their **probabilities** $q_t^{1:N}$. Unlike standard RL-based language model training, SSRL derives rewards entirely from self-supervised signals based on observations and model outputs, which are used to guide GRPO updates. (b) Reward computation in SSRL. Given the model outputs, an action likelihood evaluator (either an LLM or a model-based planner) estimates **the likelihood of the observed action** under each mental state hypothesis, and **mental priors** are estimated as the likelihood of proposed hypotheses by an LLM or set uniformly. The reward is computed as the probability-weighted log-likelihood of the observed action and mental state hypotheses minus an entropy regularization term.

Figure 2b). Through this process, *MindZero* internalizes the Bayesian inverse planning procedure in Equation (1) and enables real-time planning for proactive assistance as in Equation (2).

### 4.1. Self-Supervised RL for Mental Reasoning

Standard supervised approaches to mental reasoning rely on ground-truth mental state annotations, which are scarce and difficult to collect. Existing self-supervised methods for sequential modeling, such as next-token prediction (Bengio et al., 2003; Radford et al., 2018) and autoregressive trajectory modeling (Chen et al., 2021), emphasize forward prediction and learn by mimicking future words or actions from past context. In contrast, mental reasoning requires inverse modeling: explicitly inferring the mental state that causes the observed behavior. This capability is not explicitly learned by existing self-supervised objectives, which are optimized for prediction rather than explanation.

To bridge this gap, we formulate mental reasoning as a self-supervised reinforcement learning (SSRL) problem centered on explanatory consistency. Instead of treating actions as prediction targets, we view them as evidence. In *MindZero*, the model is rewarded not for predicting actions directly, but for generating mental state hypotheses that maximize the likelihood of user actions, thereby providing coherent explanations of agent behavior. As illustrated in Figure 2a, unlike common RL-based language model training, the reward in our SSRL method is entirely calculated via self-supervised signals from user behavior (without ground-truth mental state annotations) and model outputs. Based on this reward, we then use GRPO (Shao et al., 2024; Guo et al., 2025) to train the model, closing the self-supervised learning loop.

### 4.2. Reward Design

Formally, given a sequence of user behavior $(s_{1:t}, a_{1:t})$, we optimize a multimodal language model $Q_\theta$ to approximate the posterior of mental states $m_t$ via variational inference (Bishop, 2006). As traversing the full hypothesis space is intractable, we maximize the Evidence Lower Bound (ELBO) (Kingma & Welling, 2014). The optimization objective can be formalized as the following reward function:

$$\mathcal{J}(\theta) = \mathbb{E}_{Q_\theta}[\log(P(a_{1:t} \mid m_t, s_{1:t}) \cdot P(m_t))] + H(Q_\theta), \tag{3}$$

where the $P$ terms denote estimators of the **action likelihood** and **mental state prior** in Equation (1); and $H(Q_\theta)$ is the entropy of $Q_\theta$. In particular, the entropy term encourages exploration over mental state hypotheses and prevents premature collapse to a single mode, thereby promoting robust and diverse posterior approximations.

In practice, the model $Q_\theta(\cdot \mid s_{1:t}, a_{1:t})$ generates a finite set of $N$ mental state hypotheses $\mathcal{M}_t = \{m_t^{(1)}, \dots, m_t^{(N)}\}$, along with their normalized posterior probabilities $\mathcal{Q}_t = \{q_t^{(1)}, \dots, q_t^{(N)}\}$ such that $\sum_{i=1}^{N} q^{(i)} = 1$. We treat these $N$ candidates as the effective support of the variational posterior. Consequently, the likelihood, prior, and entropy terms in Equation (3) are computed as weighted sums:

$$R(\mathcal{M}_t, \mathcal{Q}_t) = \sum_{i=1}^{N} q_t^{(i)} [\log(P(a_{1:t} \mid m_t^{(i)}, s_{1:t}) \cdot P(m_t^{(i)}))]$$
$$- \sum_{i=1}^{N} q_t^{(i)} \log q_t^{(i)}. \tag{4}$$

**Action Likelihood.** Action likelihood measures how probable the observed actions are under a given mental state hypothesis. Specifically, $P_t^{(i)} = P(a_{1:t} \mid m_t^{(i)}, s_{1:t})$ computes the likelihood of the action sequence up to time $t$, given the observed states $s_{1:t}$ and a proposed mental state hypothesis $m_t^{(i)}$. This likelihood can be estimated using either a model-based planner (as in the GridWorld domain in Section 5.1 and 5.2) or an LLM (as in the Household domain in Section 5.3 and 5.4).

**Mental State Prior.** Mental state prior $P(m_t)$ represents the prior probabilities assigned to different mental state hypotheses $m_t$. These priors can be either uniform or non-uniform to incorporate prior knowledge from symbolic rules or LLMs, helping constrain the hypothesis space. For example, in a household environment, goals such as placing food into a dishwasher or setting the table with vastly mismatched numbers of plates and cutlery would be assigned a low prior probability. This effectively prevents the model from generating hypotheses that violate common sense at the proposal stage.

In summary, to produce hypotheses with high action likelihoods, high mental state priors, and consequently, high rewards, the proposed mental states must be explicit and meaningful for both estimators for the action likelihood and the mental state prior. This then encourages the model to learn to propose explicit and meaningful mental states through RL training. In the meantime, with the entropy bonus objective, the hypothesis distribution would remain diverse and robust. As a result, the model can learn to conduct explicit online mental reasoning without the need for ground-truth mental state annotations.

## 5. Experimental Setup

As shown in Figure 3, we systematically evaluate *MindZero* and baseline methods across four experimental settings: (1) GridWorld Question Answering (Section 5.1), (2) GridWorld Proactive Assistance (Section 5.2), (3) Household Question Answering (Section 5.3), and (4) Household Proactive Assistance (Section 5.4). The question answering settings focus on directly answering ToM-related questions about humans' mental states, whereas the assistance settings require fast, online mental reasoning about human behavior to provide proactive and accurate support. We list the evaluated models and baselines in Section 5.5.

### 5.1. GridWorld Question Answering

We adapt the *Construction* environment (Jha et al., 2024), a 2D grid world where agents navigate around obstacles (e.g., walls) and carry colored objects to different locations. Here, a human agent aims to assemble two blocks of specific colors by picking up one and moving it toward the other. The

model must infer the human's intended goal, specifically which two colored blocks the human intends to assemble, given a partial trajectory of diverse human action patterns. Beyond mental-state reasoning, the task also requires visual grounding: the model must map the question and trajectory to the correct colored blocks in the scene. This goes beyond prior ToM QA benchmarks, which are largely story-based and do not require vision-language grounding.

When training *MindZero* in the GridWorld domain, we assume a uniform prior over the reward defined in Equation (4) and use a model-based planner to estimate action likelihoods.

### 5.2. GridWorld Proactive Assistance

Using the same *Construction* environment as in Section 5.1, we define a proactive assistance task in which a human agent aims to assemble two blocks of specific colors, while a helper agent must continuously observe the human's actions, infer the intended goal, and assist in completing it more efficiently. We evaluate helping performance using speedup, which measures how much the helper accelerates the human's task completion; metric details are provided in Appendix A.2. Implementation and data generation details are provided in Appendix B.

The proactive assistance setting introduces several challenges beyond story-based evaluation: (1) reasoning must occur at *every timestep*, rather than at a single queried moment; (2) the model must generate *diverse yet plausible hypotheses from scratch*, rather than selecting from provided choices; and (3) the assistant must perform *online goal inference under ambiguity*, identifying the user's goal early enough to provide timely help, but not so early that it commits to an incorrect hypothesis. Delayed inference limits effective assistance, while premature and incorrect inference can incur *large penalties* when the assistant helps toward the wrong goal and later revises its belief.

### 5.3. Household Question Answering

We evaluate household question answering using MMToM-QA (Jin et al., 2024), a multimodal benchmark that includes questions covering the beliefs and goals of a person searching for an object (e.g., a remote controller) in a household environment. The task is challenging because it requires joint inference of both beliefs and goals with both visual and textual inputs.

For the household domain, we adopt the information fusion methods proposed by Jin et al. (2024) and (Shi et al., 2025) to combine visual and textual inputs, resulting in fused representations in text form. All methods receive the same fused information as input. When training *MindZero*, we use the same pretrained LLM to estimate both the prior

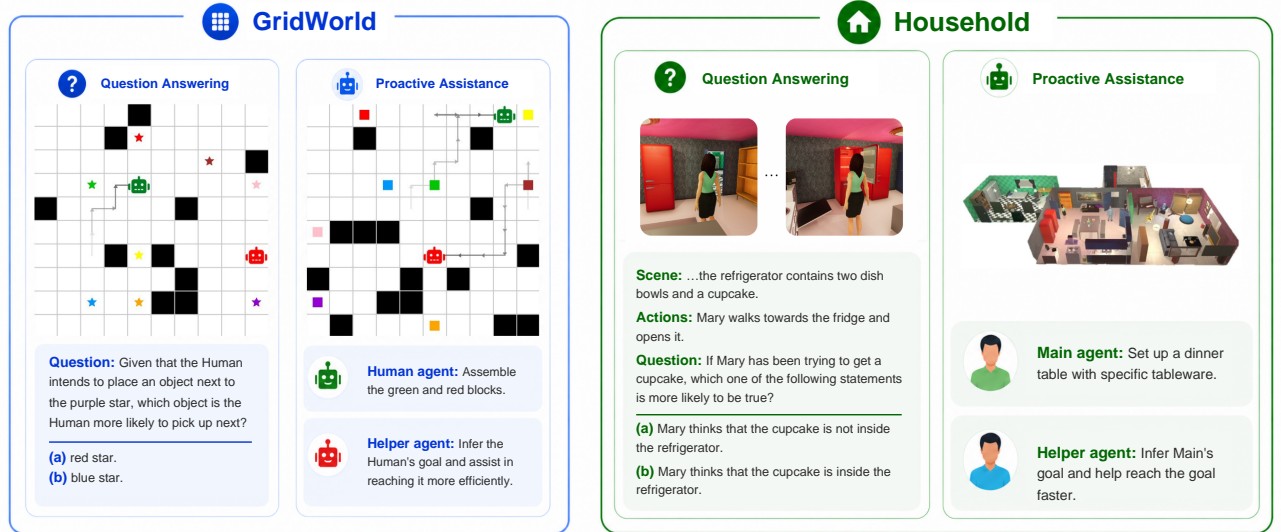

*Figure 3.* Our experimental settings for mental state reasoning and proactive assistance: (1) GridWorld Question Answering (Section 5.1); (2) GridWorld Proactive Assistance (Section 5.2); (3) Household Question Answering (Section 5.3); and (4) Household Proactive Assistance (Section 5.4).

and action likelihood terms in the reward defined in Equation (4). For the prior term, the LLM directly outputs log prior probabilities by judging whether a goal is plausible in the context of a household task. This incorporates commonsense knowledge from the pretrained LLM and helps constrain the goal space. Training data generation details are provided in Appendix C.

## 5.4. Household Proactive Assistance

We evaluate household assistance using the embodied benchmark Online Watch-And-Help (O-WAH) (Puig et al., 2023), where a helper agent observes a human's actions, infers the intended goal, and assists in completing it more efficiently in realistic household environments. In this task, the helper agent must update its goal inference based on the latest observations in an online manner. At each step, we use the uncertainty-aware helping planner proposed in Puig et al. (2023) to generate assistance actions based on the inferred goals. To evaluate generalization, we use different apartments for training and testing. To reduce variance, the results are reported as the average over 3 runs per episode. We include experiment details in Appendix C.

Besides the challenges of proactive assistance described in Section 5.2, the Household setting introduces additional difficulties: (1) a much larger state, action, and goal space (e.g., uncertainty over which objects are needed, how many are required, and their target locations); (2) partial observability, whereas GridWorld is fully observable; and (3) significantly longer episode horizons.

## 5.5. Models and Baselines

We compare *MindZero* against the following baselines:

- **Base models:** For the GridWorld domain (Section 5.1–5.2), we use the open-weight multimodal models Qwen3-VL-4B and Qwen3-VL-8B (Yang et al., 2025). For the Household domain (Section 5.3–5.4), we use the open-weight language models Llama-3.1-8B, Llama-3.2-3B (Dubey et al., 2024), and Qwen3-4B (Yang et al., 2025), using fused textual inputs.

- **Large models:** Additionally, we evaluate Qwen3-235B-A22B, GPT-5.2, and Gemini-3 as zero-shot performance of large models. For question answering, we report results with both the thinking and non-thinking version of the models. For proactive assistance, we report only the non-thinking results, as it requires models to make decisions in the real time.

- **Test-time scaling methods:** We evaluate *ThoughtTracing* (Kim et al., 2025), a test-time reasoning approach for mental-state tracking that maintains and updates multiple hypotheses, and *AutoToM* (Zhang et al., 2025), a model-based method for automated agent modeling. Both are instantiated with the open-source base models listed above. We do not evaluate them in the Proactive Assistance domains due to their slow inference speed, which limits real-time applicability. As they do not support visual inputs, we provide textual transcripts of GridWorld observations. We describe implementation details in Appendix E.

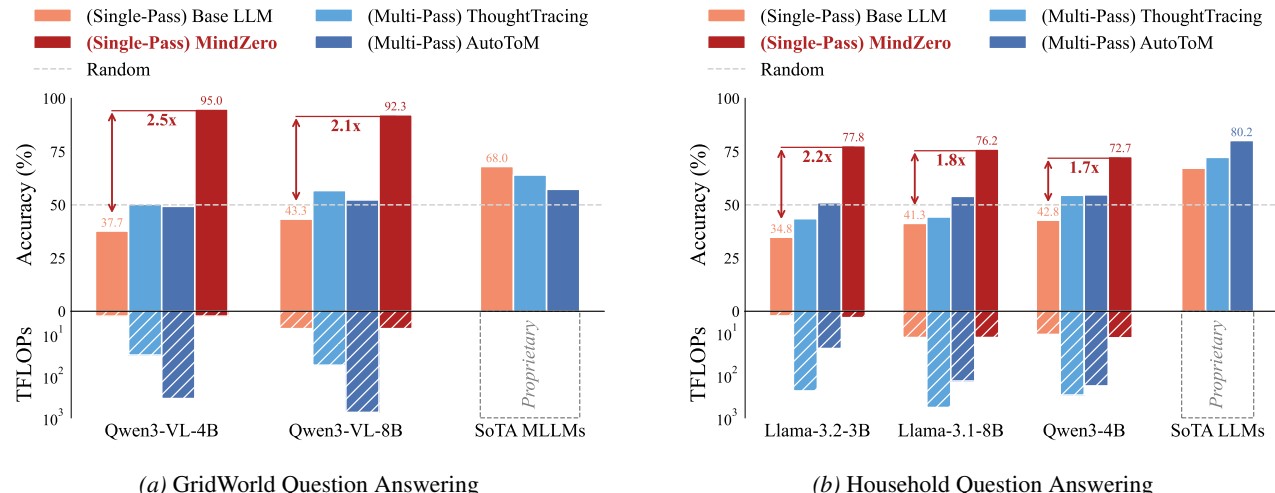

*(a)* GridWorld Question Answering
        *(b)* Household Question Answering

*Figure 4.* Question answering results of **MindZero** and baselines on (a) GridWorld and (b) Household domains. *MindZero* achieves a 1.7–2.5× accuracy (solid bars) gain across different base models with negligible additional inference cost (hatched bars), and consistently outperforms all test-time scaling baselines in both accuracy and efficiency. Full results are shown in Table 4.

For a fair comparison, we evaluate *MindZero* using the same open-source base models described above.

## 6. Experimental Results

### 6.1. Overall Results

**Question Answering**  As shown in Figure 4, *MindZero* consistently outperforms pretrained and test-time scaling baselines in both GridWorld QA (Figure 4a) and Household QA (Figure 4b), while maintaining low inference cost.

In GridWorld QA, *MindZero* achieves the best accuracy among all methods with both Qwen3-VL-4B and Qwen3-VL-8B, substantially improving over their base models and delivering a 2.1–2.5× accuracy gain.

In Household QA, *MindZero* likewise achieves strong performance across all base models, with *MindZero* w/ Llama-3.2-3B attaining the highest accuracy among open-weight and test-time scaling methods and remaining competitive with the best proprietary systems despite minimal inference cost. Compared with *ThoughtTracing* and *AutoToM*, which require substantially more test-time computation, *MindZero* delivers a clearly better accuracy-efficiency trade-off, even when those methods use much larger backend models.

**Proactive Assistance**  As shown in Table 1, *MindZero* achieves the best performance among all and yields substantial gains from base models in task completion speed in both GridWorld Proactive Assistance (Table 1a) and Household Proactive Assistance (Table 1b), where all baselines provide little to no speedup.

In GridWorld Proactive Assistance, *MindZero* achieves 23.0% and 24.5% speedup with Qwen3-VL-4B and Qwen3-

VL-8B, respectively. In contrast, GPT-5.2 and Gemini-3-Flash yield no speedup, as their goal predictions change constantly, causing the agent's actions to become unstable (i.e., frequently changing directions). As a result, the agent fails to pick up an object before the task ends.

In Household Proactive Assistance, *MindZero* with Qwen3-4B achieves a best speedup of 19.1%, significantly higher than the strongest baseline with the least inference cost. A notable exception is *MindZero* with Llama-3.2-3B, which does not show a significant gain over its base model. This is because it cannot produce goal hypotheses in the required format, we first fine-tune it on generations sampled from the pretrained Llama-3.1-8B before RL training, avoiding any reliance on ground-truth or pseudo labels. However, while this warm-up teaches the correct format, the relatively low quality of the sampled generations appears to be memorized as well, introducing a bias that ultimately suppresses the expected improvement.

### 6.2. Online Goal Inference Dynamics

Figure 5 shows the accuracy of online goal inference as task progress increases in both GridWorld and Household Proactive Assistance. In both settings, *MindZero* steadily improves its goal prediction over time, indicating that it can effectively accumulate evidence from ongoing interaction and refine its belief about the user's objective. In GridWorld (Figure 5a), *MindZero* is the only method whose accuracy rises substantially as the task unfolds, eventually reaching a strong level. In contrast, all baselines remain very low for most of the trajectory and only increase in accuracy near the end, making effective assistance difficult. In Household (Figure 5b), *MindZero* again achieves the

*Table 1.* Proactive assistance results of **MindZero**, base models, and large models on (a) Gridworld and (b) Household domains. Best results are shown in **bold**. * indicate models that cannot generate goal hypotheses in the correct format at all, and need to be finetuned to follow output format before the RL training.

<table>
<tr><td colspan="3" align="center">*(a)* Gridworld Proactive Assistance</td></tr>
<tr><td>Method</td><td>Speedup ↑</td><td>TFLOPs ↓</td></tr>
<tr><td>Random Goal</td><td>0.0</td><td>N/A</td></tr>
<tr><td>Base Models</td><td></td><td></td></tr>
<tr><td>Qwen3-VL-4B</td><td>1.4</td><td>**151.7**</td></tr>
<tr><td>Qwen3-VL-8B</td><td>-0.1</td><td>295.2</td></tr>
<tr><td>Large Models</td><td></td><td></td></tr>
<tr><td>Qwen3-VL-235B-A22B</td><td>1.0</td><td>808.6</td></tr>
<tr><td>GPT-5.2</td><td>0.0</td><td>Proprietary</td></tr>
<tr><td>Gemini-3-Flash</td><td>0.0</td><td>Proprietary</td></tr>
<tr><td>*MindZero (Ours)*</td><td></td><td></td></tr>
<tr><td>w/ Qwen3-VL-4B</td><td>23.0</td><td>161.4</td></tr>
<tr><td>w/ Qwen3-VL-8B</td><td>**24.5**</td><td>291.8</td></tr>
</table>

<table>
<tr><td colspan="3" align="center">*(b)* Household Proactive Assistance</td></tr>
<tr><td>Method</td><td>Speedup ↑</td><td>TFLOPs ↓</td></tr>
<tr><td>Random Goal</td><td>-2.2</td><td>N/A</td></tr>
<tr><td>Base Models</td><td></td><td></td></tr>
<tr><td>Llama-3.2-3B*</td><td>2.3</td><td>244.3</td></tr>
<tr><td>Llama-3.1-8B</td><td>1.7</td><td>656.1</td></tr>
<tr><td>Qwen3-4B</td><td>2.3</td><td>213.1</td></tr>
<tr><td>Large Models</td><td></td><td></td></tr>
<tr><td>Qwen3-235B-A22B</td><td>12.3</td><td>1101.6</td></tr>
<tr><td>GPT-5.2</td><td>9.4</td><td>Proprietary</td></tr>
<tr><td>Gemini-3-Flash</td><td>17.7</td><td>Proprietary</td></tr>
<tr><td>*MindZero (Ours)*</td><td></td><td></td></tr>
<tr><td>w/ Llama-3.2-3B*</td><td>4.3</td><td>235.1</td></tr>
<tr><td>w/ Llama-3.1-8B</td><td>17.4</td><td>608.4</td></tr>
<tr><td>w/ Qwen3-4B</td><td>**19.1**</td><td>**201.2**</td></tr>
</table>

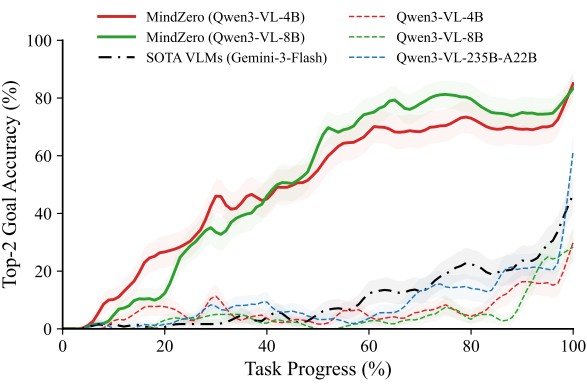

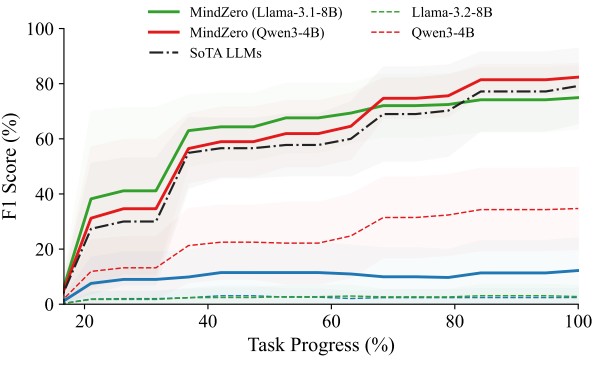

*(a)* GridWorld Proactive Assistance

*(b)* Household Proactive Assistance

*Figure 5.* Goal accuracy or F1 score for online goal inference versus task progress across (a) GridWorld and (b) Household proactive assistance. *MindZero*'s (bold solid curves) predicted goal steadily improves in accuracy over time and reaches a strong level, while most baselines (dashed curves) remain much lower or improve more slowly.

strongest performance, with prediction accuracy increasing consistently, significantly outperforming base models and matching much larger pretrained models. These results suggest that accurate and stable online goal inference is a key reason why *MindZero* can deliver effective proactive assistance.

### 6.3. Ablation Study

To understand the key components driving *MindZero*'s performance, we conduct comprehensive ablation studies on Qwen3-4B, as shown in Table 2. We examine three critical design choices: prior modeling, multiple hypotheses,

and entropy bonus. All experiments use the same training configuration as our main experiments.

**Explicit Prior Modeling**   In the household environment, humans are assumed to pursue a set of predefined goal types, such as setting up the dinner table or putting dishes in the dishwasher. We explicitly require an LLM to check whether each goal hypothesis is reasonable. For example, putting an apple into the dishwasher will be assigned a very low score. This constraint is key to generating plausible hypotheses and prevents reward hacking of action likelihood, e.g., including every possible item in the goal yields a high

*Table 2.* Ablation on Household Proactive Assistance using Qwen3-4B.

| # | Method | Speedup ↑ | TFLOPs ↓ |
|---|--------|-----------|----------|
| I | *MindZero* | **19.1** | 201.2 |
| II | w/o prior modeling | 17.0 | 200.5 |
| III | w/o multiple hypotheses | 10.3 | **132.6** |
| IV | w/o entropy bonus | 5.2 | 245.1 |

action-likelihood score but a low prior score. Compared to the full model (Row I), the speedup drops by 2.1% without explicit prior modeling (Row II).

**Multiple Hypotheses** Maintaining a set of mental state hypotheses is important for capturing the uncertainty of understanding human behavior. For example, in the early stage of an episode, the assistant can only observe a limited human behavior, thus each hypothesis remains ambiguous and carries low confidence. Relying on a single estimation would lead to premature commitment to a potentially incorrect goal. By tracking a beam of hypotheses, the system can defer the decision until sufficient evidence is accumulated. Compared to the full model (Row I), the speedup drops for 8.8% comparing to generating a single most possible mental state (Row III). Accordingly, the token usage is the least.

**Entropy Bonus** Hypothesis distribution often suffers from mode collapse, where the model becomes overconfident in a single prediction too early. To mitigate this, the entropy regularization term in Equation (3) encourages the diversity of the hypothesis space. This bonus penalizes overly peaked distributions and ensures the model retains alternative possibilities during reasoning. Compared to the full model (Row I), the speedup drops for 13.9% without the entropy bonus (Row IV).

### 6.4. Human Experiment

To evaluate whether *MindZero* can support real users, we conducted a human experiment in the Household Proactive Assistance domain. Participants acted as the main agent and completed four household tasks from our test set. We recruited 12 participants from Johns Hopkins University. The study was approved by the JHU institutional review board.

**Experimental Setup.** We compare four settings: a Single Human without assistance, and assistance with Qwen3-4B, with *MindZero* trained from Qwen3-4B, and with Gemini-3-Flash. The Single Human setting serves as the reference for computing speedup. All assisted settings use the same helper-agent pipeline as in the Household Proactive Assistance experiments, varying only the mental inference model.

**Results.** The pretrained Qwen3-4B model yields only a marginal speedup of 2.6%. In contrast, *MindZero* trained from Qwen3-4B achieves a speedup of 19.7% (standard error 6.3%), a substantial improvement over the same Qwen3-4B backbone. Gemini-3-Flash achieves a speedup of 23.4% (standard error 6.4%). Although Gemini-3-Flash attains a slightly higher mean speedup, the difference between Gemini-3-Flash and *MindZero* is not statistically significant under a paired t-test on speedup ($p = 0.24$), consistent with the results in Section 6.

These results show that *MindZero* transfers to real human behavior and provides effective assistance. *MindZero* reaches performance comparable to Gemini-3-Flash while using a small open-weight model, making it easier to deploy locally and more cost-effective for large-scale assistance.

## 7. Conclusion

We introduced *MindZero*, a self-supervised reinforcement learning framework for training multimodal language models to perform robust and efficient online Theory of Mind reasoning without relying on mental state annotations. By rewarding hypotheses that best explain observed behavior, *MindZero* enables models to internalize the deliberative structure of model-based ToM while retaining the speed of single-pass inference. Extensive evaluations across question answering and proactive assistance tasks demonstrate that *MindZero* achieves strong robustness and uncertainty tracking comparable to explicit model-based methods, while substantially reducing computational cost. These results show that mental reasoning can be learned as a self-supervised skill grounded in behavioral evidence, bridging the long-standing gap between interpretability, robustness, and efficiency in ToM modeling. We believe *MindZero* provides a promising foundation for scalable, real-world assistive agents that can continuously reason about human intentions and adapt to dynamic environments.

**Limitations and Future Work.** Our current *MindZero* framework does not model recursive reasoning between multiple agents. Additionally, as the input sequence length increases, the required input token length for the model will increase accordingly. In the future, we intend to expand *MindZero* to incorporate multi-agent recursive mental reasoning into the training process. We also plan to develop a more efficient model structure to address the challenge of long input sequences.

## Impact Statement

This paper presents work aimed at advancing the field of machine learning by developing more robust and efficient methods for online Theory of Mind reasoning in assistive AI systems. By enabling models to infer human intentions and uncertainty from behavior without relying on explicit annotations, our approach has the potential to enhance the reliability, responsiveness, and scalability of AI agents in real-world applications such as household assistance, digital services, and human–computer interaction. These advances may contribute to more helpful, adaptive, and accessible technologies that better align with users' needs and preferences, thereby improving user experience and productivity.

At the same time, enhanced mental reasoning capabilities may raise ethical considerations. Systems that more accurately model human intentions and beliefs may be misused for manipulation, surveillance, or unwanted behavioral profiling if deployed without appropriate safeguards. Moreover, errors in inferred mental states could result in inappropriate assistance, reduced user autonomy, or the reinforcement of existing biases present in behavioral data. We emphasize that responsible use requires transparency, user consent, and careful evaluation in real-world settings. We hope this research encourages further discussion on the ethical development and deployment of human-centered AI systems and supports future work on fairness, accountability, and privacy-preserving mental reasoning models.

## Author Contributions

Shunchi Zhang conceived the idea and developed it into the present work; he carried out the main environment setup, data processing, model training, and evaluation, including the extensive exploratory experiments and the core experimental results reported in the paper. Jin Lu conducted a large number of additional experiments, primarily baselines and supplementary studies; he also independently performed the human study, contributed to the early-stage exploration of the GridWorld experiments, and carried out exploratory work on web assistance that informed the final design. Chuanyang Jin contributed to paper writing and figure design. Yichao Zhou implemented the GridWorld environment setup, data processing, model training, and evaluation, under Shunchi Zhang's assistance. Zhining Zhang contributed the AutoToM-related experiments. Tianmin Shu provided overall research direction and weekly guidance and contributed to the paper revision. All authors contributed to the paper writing.

## Acknowledgement

This work is supported by a grant from Amazon. Chuanyang Jin is supported by the Amazon AI PhD Fellowship.

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

# A. *MindZero* Implementation Details

## A.1. Model Training

All *MindZero* models are trained with standard GRPO in the VeRL framework using 4×H100 GPUs. For Household domain, we additionally serve a reward model Qwen3-235B-A22B-FP8 model with vLLM using 4×H100 GPUs. We use 32 rollout samples per prompt as the hypothesis proposal set, a rollout batch size of 32, a global batch size of 8, and train for 20 epochs with AdamW in bf16. The main optimization hyperparameters are a learning rate of $1 \times 10^{-6}$, weight decay of $1 \times 10^{-2}$, a max grad norm of 1.0, and a KL coefficient of $1 \times 10^{-2}$. Detailed configurations are open-sourced at https://github.com/SCAI-JHU/MindZero/tree/main/configs.

## A.2. Evaluation Metrics

**Speedup in Proactive Assistance.** We measure collaborative efficiency using the *speedup* metric:
$$\text{speedup} = \frac{T_{\text{human}}}{T_{\text{collab}}} - 1 \tag{5}$$
where $T_{\text{human}}$ denotes the time required when the helper remains stationary, and $T_{\text{collab}}$ denotes the time taken with active assistance.

**Inference Cost.** We report the inference cost in terms of floating point operations (FLOPs) using the approximation
$$\text{FLOPs} = 2 \times P_{\text{active}} \times N_{\text{tokens}}$$
$$\frac{\text{FLOPs}}{\text{Trillion}} = 2 \times \frac{P_{\text{active}}}{\text{Billion}} \times N_{\text{tokens}} \times \frac{1}{1000}, \tag{6}$$
where $P_{\text{active}}$ denotes the active parameter count and $N_{\text{tokens}}$ represents the total number of processed tokens (Kaplan et al., 2020).

## A.3. Prompt Examples

We use the same instruction but different context inputs for every task. Examples are shown in Figure 6-9 for task context.

For reward evaluation in Household domain, we adopt similar prompts in AutoToM (Zhang et al., 2025).

All prompts are open-sourced at https://github.com/SCAI-JHU/MindZero/tree/main/prompts, and datasets are open-sourced at https://huggingface.co/datasets/SCAI-JHU/MindZero.

# B. GridWorld Experiments

We provide the experimental details of our GridWorld Question Answering (Section 5.1) and Proactive Assistance (Section 5.2) experiments.

## B.1. Environment Setup

We randomly generate episodes in a $10 \times 10$ grid world containing $U(0, 20)$ obstacles and 8 uniquely colored and shaped objects. To ensure task complexity, generated episodes are filtered to guarantee sufficient trajectory length and goal ambiguity. The resulting dataset comprises both rendered visual observations and detailed textual descriptions of the environment rules.

All environments and agents accept explicit seeds. We store environment configurations, initial states, and full action histories to reproduce any episode or visualization.

## B.2. Data Generation

**Question Answering** We formulate the QA task using binary-choice questions with grounded natural language descriptions. For each episode, we generate three distinct types of queries to test different aspects of social reasoning:

- **Type 1 & 2 (Pre-Pick):** Sampled at timesteps before the human picks up an object. These questions query the model's ability to infer the intended object to be picked (given the placement goal) or the overall goal configuration.

- **Type 3 (Post-Pick):** Sampled at timesteps after the human is holding an object. These questions query the intended

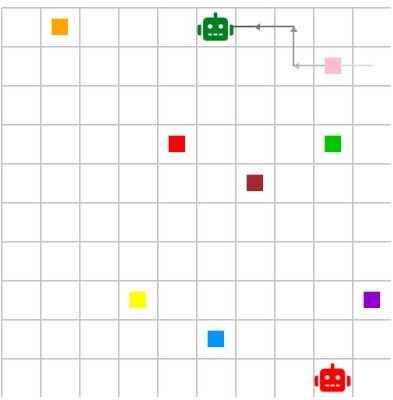

```
// context
<image> You are a helper agent in a GridWorld environment. You are the red robot, and the Human is the green robot. There are
multiple objects: brown square, pink square, red square, blue square, green square, yellow square, orange square, and purple
square. The Human's goal is to place two of the objects next to each other. The Human can move up, down, left, right, or stay,
and can pick up an object when standing on it and not holding one, and can put down an object when holding one and the cell is
empty. The Human's action trajectory so far is shown in the image. Given that the Human intends to place an object next to the
yellow square, which object is the Human more likely to pick up next? (a) orange square. (b) green square.

// instruction
Please respond with only a single lower case letter a or b.
```

*Figure 6.* A prompt example for GridWorld Question Answering.

placement target given the currently held object.

We utilize 800 episodes (2,400 questions) for training and 100 episodes (300 questions) for evaluation.

**Proactive Assistance**   For proactive assistance, the model is required to propose a full probability distribution over the $N$ candidate goal pairs at each timestep, enabling real-time intent inference without explicit questioning. We use $N = 2$ in the experiments. To enhance visual grounding and standardize the goal representations, we impose a strict structural constraint on our model's output. Specifically, the model is instructed that the human agent consistently prioritizes interacting with the nearest object first. Consequently, within each predicted goal hypothesis, the objects must be strictly ordered based on their initial proximity to the human's starting position (i.e., the closer object is explicitly designated as the first object, and the further one as the second). This structured output formulation provides a stronger spatial inductive bias compared to the unconstrained inference prompts used for the pretrained baselines.

We employ 1000 unlabeled episodes, unrolled into individual timesteps, for training the stepwise inference model. Evaluation is performed on a separate set of 20 randomly sampled episodes to assess online assistance performance.

### B.3. Agent Policies

**Helping Planner**   The helper assists the human by maintaining a goal distribution $B = \{(g_i, p_i)\}$ over paired goals. It selects actions using a Boltzmann policy based on the probability-weighted expected return: $Q(a) = \sum_i p_i \cdot V(a \mid g_i)$. The policy is designed to be complementary: it predicts which target the human will prioritize (typically the closer one) and aims for the other. To ensure smooth collaboration, the helper follows heuristic rules to yield to the human, avoid blocking paths, and prevent deadlocks.

**Simulated Human Planner**   The human agent employs a goal-directed planner based on shortest-path distances, operating sequentially by acquiring the proximal target and transporting it to a position adjacent to the distal target. Actions are sampled via a Boltzmann policy with temperature $\tau = 0.01$, subject to logical constraints (e.g., mandatory object interactions). To simulate physical load constraints in the proactive assistance task, the human adheres to an alternating "move-then-pause" pattern when carrying an object. Furthermore, to mimic realistic human stochasticity and enhance trajectory diversity, we introduce a randomness factor of $0.15$ during evaluation, where the agent takes a random action with $15\%$ probability. To

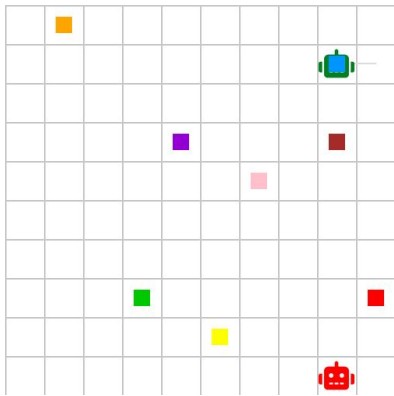

```
// context
<image> You are a helper agent in a GridWorld environment. You are the red robot, and the Human is the green robot. There are
multiple objects: pink square, blue square, purple square, yellow square, brown square, green square, orange square, and red
square. The Human's goal is to place two of the objects next to each other. The Human can move up, down, left, right, or stay,
and can pick up an object when standing on it and not holding one, and can put down an object when holding one and the cell is
empty. The Human's action trajectory so far is shown in the image. Please propose a probability distribution that includes 2
candidate paired goals and their probabilities. Your response should include the probability distribution formatted according to
this JSON schema: {"$defs": {"GoalParticle": {"properties": {"object1": {"$ref": "#/$defs/Object"}, "object2": {"$ref": "#/$defs/
Object"}, "p": {"description": "Probability of the goal proposal", "maximum": 1, "minimum": 0, "title": "P", "type": "number"}},
"required": ["object1", "object2", "p"], "title": "GoalParticle", "type": "object"}, "Object": {"properties": {"color": {"title":
"Color", "type": "string"}, "shape": {"title": "Shape", "type": "string"}}, "required": ["color", "shape"], "title": "Object", "
type": "object"}}, "properties": {"particles": {"items": {"$ref": "#/$defs/GoalParticle"}, "title": "Particles", "type": "array
"}}, "required": ["particles"], "title": "GoalParticles", "type": "object"}.

// instruction
Note that the Human (green robot) consistently prioritizes picking up the object closest to its initial starting position first,
subsequently placing it next to the object that was initially further away. In your JSON response, ensure that for every
GoalParticle, object1 is strictly the object closer to the Human (green robot)'s starting position, and object2 is the object
further from it.
Please output the minified JSON.
```

*Figure 7.* A prompt example for GridWorld Proactive Assistance.

```
// context
What's inside the apartment: There is a kitchen and a bathroom and a bedroom and a living room.
four kitchen cabinets and a stove and a refrigerator and a microwave and a kitchen table are in the kitchen. a condiment bottle
is on the fourth kitchen cabinet. a dish bowl and two wine glasses and an apple are on the first kitchen cabinet. a dish bowl and
 a bottle of wine and a condiment bottle and a wine glass are on the third kitchen cabinet. There is nothing inside the stove. a
plate and a cupcake and a bottle of wine and a dish bowl are inside the refrigerator. a salmon is inside the microwave.
a bathroom cabinet is in the bathroom. There is nothing inside the bathroom cabinet.
a coffee table and a desk are in the bedroom.
a coffee table and a cabinet and a desk and a sofa are in the living room. a water glass and a book are on the coffee table. two
cupcakes and two dish bowls and a remote control and a wine glass are inside the cabinet.
Actions taken by Mary: Mary is inside the bedroom. Mary walks towards the kitchen.
Question: If Mary has been trying to get a dish bowl, which one of the following statements is more likely to be true? (a) Mary
thinks that the dish bowl is inside the kitchen. (b) Mary thinks that the dish bowl is not inside the kitchen. Please respond
with either a or b.

// instruction
You FIRST think about the reasoning process as an internal monologue and then provide the final answer. The reasoning process
MUST BE enclosed within <thinking> </thinking> tags. The final answer MUST BE put in \boxed{}.
```

*Figure 8.* A prompt example for Household Question Answering.

account for the stochasticity of our helping planner and simulated human planner, we evaluate the GridWorld proactive
assistance task across three random seeds (10, 20, and 30) and report the averaged results.

```
// context
Human has been working on a task of moving some objects to a target location. The task type can only be one of the following:
setting up a table, putting something in the dishwasher, putting something in the fridge, preparing food, or watching TV.

Your are a helpful assistant. In order to help human, please propose multiple hypotheses of [human's overall goal] (including
both finished and potential future subgoals), base on the following information:

[current state]
The apartment has 4 rooms: bathroom, bedroom, kitchen, livingroom.

The bathroom has 1 bathroomcabinet.

The bedroom has 1 coffeetable.
- The coffeetable supports 1 wineglass, 1 plate.

The kitchen has 1 fridge, 4 kitchencabinet, 1 kitchentable, 1 microwave, 1 stove.
- The fridge contains 1 plate, 2 cupcake, 1 salmon, 1 pudding.
- The kitchencabinet contains 1 apple, 3 cutleryfork.
- The kitchencabinet contains 1 wineglass, 1 cutleryfork.
- The kitchencabinet contains 1 wineglass, 1 cutleryfork.
- The kitchencabinet contains 2 condimentbottle.
- The microwave contains 1 condimentbottle, 1 salmon.
- The stove contains 1 salmon, 1 cupcake.

The livingroom has 1 cabinet, 1 coffeetable.
- The cabinet contains 1 remotecontrol, 1 cupcake, 1 wineglass.
- The coffeetable supports 1 plate, 1 remotecontrol.

Human is in the kitchen.
Human is close to 4 wallpictureframe, 1 salmon, 2 condimentbottle, 1 microwave, 1 wallphone, 6 bellpepper, 3 kitchencounterdrawer
, 1 dishbowl, 1 clock, 1 lightswitch, 1 pudding, 1 cutleryknife, 1 plate, 1 fridge, 1 powersocket, 1 book, 1 bench, 1 sink, 1
kitchencounter, 1 kitchencabinet, 1 rug.
Human is holding nothing.

[key action history]
Human has not taken any key action yet.

[human's next action]
Human walks towards the kitchencabinet

Hints:
- The task type is constant and the target location is unique, i.e., human will be consistently doing the same task (setting up a
  table, putting something in the dishwasher, putting something in the fridge, preparing food, or watching TV) and put all objects
  to the same location.
- Please propose diverse goals in both object type and count.

Output Requirements:
Please provide a probability distribution over n=10 hypotheses of [human's overall goal] (including both finished and potential
future subgoals).
Your response should include the probability distribution formatted according to this JSON schema: {'$defs': {'GoalParticle': {'
properties': {'task_name': {'enum': ['prepare_food', 'put_dishwasher', 'put_fridge', 'setup_table', 'watch_tv'], 'title': 'Task
Name', 'type': 'string'}, 'objects': {'items': {'$ref': '#/$defs/Object'}, 'minItems': 1, 'title': 'Objects', 'type': 'array'}, '
target': {'$ref': '#/$defs/Target'}, 'p': {'description': 'Probability of the goal proposal', 'maximum': 1, 'minimum': 0, 'title
': 'P', 'type': 'number'}}, 'required': ['task_name', 'objects', 'target', 'p'], 'title': 'GoalParticle', 'type': 'object'}, '
Object': {'properties': {'type': {'enum': ['apple', 'chips', 'condimentbottle', 'cupcake', 'cutleryfork', 'plate', 'pudding', '
remotecontrol', 'salmon', 'waterglass', 'wineglass'], 'title': 'Type', 'type': 'string'}, 'count': {'minimum': 1, 'title': 'Count
', 'type': 'integer'}}, 'required': ['type', 'count'], 'title': 'Object', 'type': 'object'}, 'Target': {'properties': {'type': {'
enum': ['coffeetable', 'dishwasher', 'fridge', 'kitchentable', 'stove'], 'title': 'Type', 'type': 'string'}}, 'required': ['type
'], 'title': 'Target', 'type': 'object'}}, 'properties': {'particles': {'items': {'$ref': '#/$defs/GoalParticle'}, 'title': '
Particles', 'type': 'array'}}, 'required': ['particles'], 'title': 'GoalParticles', 'type': 'object'}

// instruction
Please output the minified JSON.
```

*Figure 9.* A prompt example for Household Proactive Assistance.

# C. Household Experiments

We provide the experimental details of our Household Question Answering (Section 5.3) and Proactive Assistance (Section 5.4) experiments.

## C.1. Environment Setup

We use `VirualHome` (Puig et al., 2021) v2.2.4 as household simulator, where agent policies are implemented by a goal-conditioned MCTS planner. For online goal inference, following AutoToM (Zhang et al., 2025), we use Sequential Monte Carlo algorithm to maintain the goal hypotheses over time.

## C.2. Data Generation

**Question Answering**    We use the MMToM-QA (Jin et al., 2024) training set to construct training data for *MindZero*. Since the test questions use binary choices, valid hypotheses may often lie outside the provided candidate set. To better match this format, we apply hypothesis filtering to construct binary options instead of sampling from the full hypothesis space. For goal-related questions, we form choices by pairing a randomly sampled observed object with an unobserved one. For belief-related questions, we sample an unobserved object–container pair to create a binary verification task. Applying this filtering strategy to the 953 training episodes yields a final dataset of 4,866 examples.

**Proactive Assistance**    Following the standard setting of `VirualHome` (Puig et al., 2021), we use Apartment #0, #1, #2, #3, and #5 for training data generation, and Apartment #3 and #6 for testing data generation. We generate 20 episodes (968 timesteps) for training and 16 for testing, evenly distributed across four task types: setting up a table, loading the fridge, preparing food, and loading the dishwasher.

# D. Human Experiment

We recruited 12 Johns Hopkins University students, including undergraduate, master's, and Ph.D. students. The pool included 5 male and 7 female participants. All participants were at least 18 years old and able to operate a computer interface. The study was approved by the Institutional Review Board (IRB). Prior to participation, each participant reviewed and signed an informed consent form. Participation was voluntary, and participants could withdraw from the study at any time.

Each study session took approximately 60 minutes. Participants completed household tasks in a simulated apartment environment using a computer interface, as shown in Figure 10. During the task, the system recorded task-related interaction logs.

While Section 6.4 reports speedup averaged across tasks, we present per-task results in Table 3. Across all four tasks, MindZero trained from Qwen3-4B yields a positive speedup, whereas the pretrained Qwen3-4B model produces a negative speedup on Tasks 5 and 13, indicating that the same backbone without our training may even slow the human down. The per-task gap between *MindZero* and Gemini-3-Flash is small and varies in sign, consistent with the absence of a statistically significant difference between the two on aggregate speedup.

*Table 3.* Human experiment results in the Household Proactive Assistance domain. We report average task-completion steps for each condition and the corresponding speedup over the Single Human setting. We use *MindZero* w/ Qwen3-4B as the base model.

| Task ID | Average Steps | | | | Speedup (%) | | |
|---|---|---|---|---|---|---|---|
| | Qwen3-4B | *MindZero* | Gemini-3-Flash | Single Human | Qwen3-4B | *MindZero* | Gemini-3-Flash |
| 3 | 56 | 51 | 50 | 70 | 23.67 | 37.50 | 38.41 |
| 5 | 58 | 44 | 39 | 47 | -18.50 | 7.63 | 21.55 |
| 8 | 43 | 41 | 44 | 47 | 9.23 | 15.45 | 7.58 |
| 13 | 119 | 97 | 91 | 114 | -3.92 | 18.28 | 26.10 |
| Average | – | – | – | – | 2.62 | 19.70 | 23.40 |
| Standard Error | – | – | – | – | 9.00 | 6.30 | 6.40 |

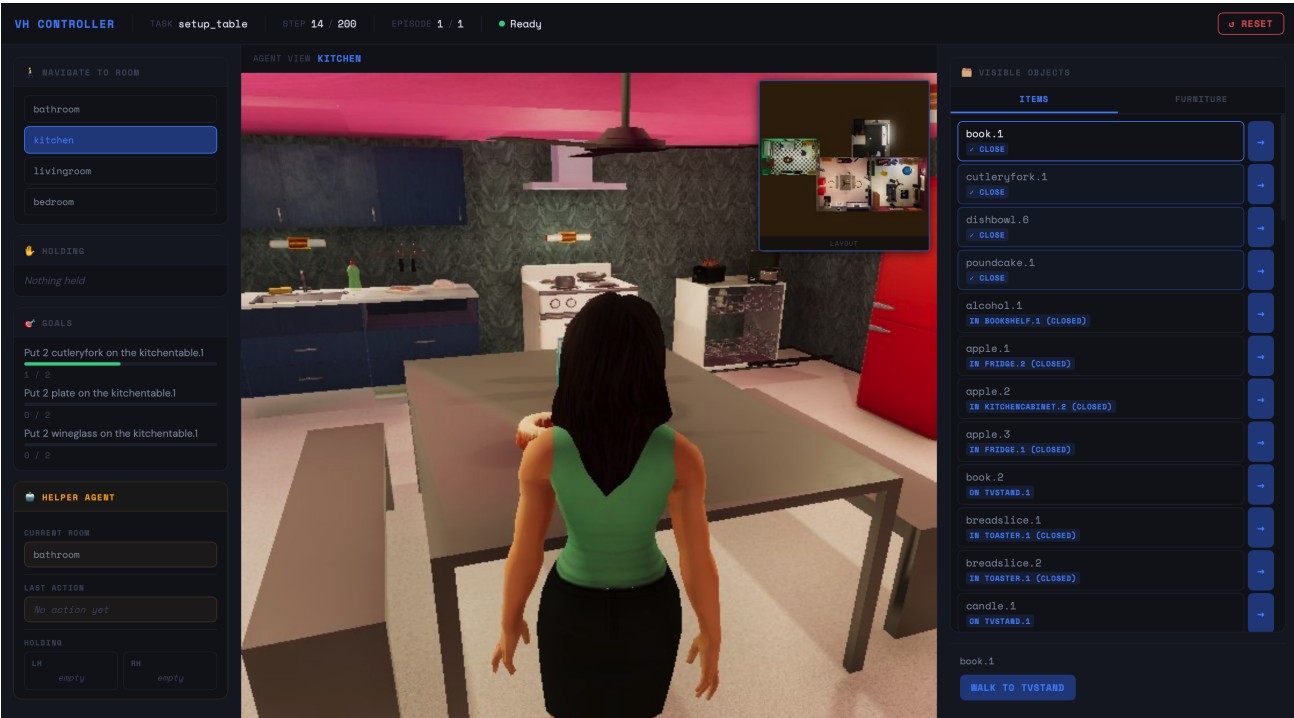

*Figure 10.* Human experiment interface for the Household Proactive Assistance domain. The header reports the task, step budget, and episode; the left panel lets the participant navigate rooms and shows holding status, goal progress, and the helper agent's state. The center renders the agent's view of the current room with an inset household map, and the right panel lists all visible objects with their spatial relations and open/closed states alongside the contextual action for the selected object.

# E. Test-Time Scaling Methods

## E.1. *ThoughtTracing*

For the Household Question Answering task, we evaluate *ThoughtTracing* using the original implementation, without any modifications to the codebase, including the prompts. In contrast to the evaluation protocol reported in the original work, we conduct our testing on the complete, unmodified set of 600 test instances to ensure a fair comparison with other baselines and our main experiments. For the GridWorld Question Answering task, which was not explored in the original work, we introduce only the necessary environment-specific modifications. As *ThoughtTracing* does not support direct visual input, we augment each question with explicit coordinate representations alongside an ASCII map of the environment. This adaptation ensures that all essential visual information required for reasoning is preserved.

## E.2. *AutoToM*

We evaluate *AutoToM* across multiple backend models using the original implementation, without any modifications to the codebase, including the prompts. Due to the limited instruction-following capabilities of smaller models (e.g., Llama-3.2-3B), parsing errors may occur. When such errors arise, we adopt a uniform distribution as the inference result of *AutoToM* to ensure a fair comparison.

## E.3. Textual Transcripts

Specifically, for GridWorld Question Answering, as both *ThoughtTracing* and *AutoToM* do not support multimodal inputs, we use textual transcripts to evaluate the performance. See an example in Figure 11.

```
You are a helper agent in a GridWorld environment. You are the red robot, and the Human is the green robot. There are multiple
objects: brown star, orange star, yellow star, pink star, green star, red star, purple star, and blue star. The Human's goal is
to place two of the objects next to each other. The Human can move up, down, left, right, or stay, and can pick up an object when
 standing on it and not holding one, and can put down an object when holding one and the cell is empty. The Human's action
trajectory so far is shown in the image.

State and trajectory details:
Agents:
- Human pos: (6, 1)
- Helper pos: (0, 0)
Obstacles:
[]
Objects (by label):
{'brown star': (5, 9), 'orange star': (8, 2), 'yellow star': (4, 0), 'pink star': (9, 1), 'green star': (5, 5), 'red star': (3,
3), 'purple star': (3, 7), 'blue star': (2, 8)}
Action deltas (dx, dy):
{'up': (0, 1), 'down': (0, -1), 'left': (-1, 0), 'right': (1, 0), 'stay': (0, 0), 'pick': (0, 0), 'put': (0, 0)}
Action trajectory (human, name + delta):
t=1: left (-1, 0); t=2: down (0, -1); t=3: down (0, -1); t=4: left (-1, 0)
Action trajectory (human positions):
t=1: (7, 3); t=2: (7, 2); t=3: (7, 1); t=4: (6, 1)
ASCII state:
Step 4
    .    .    .    .    .    0    .    .    .    .
    .    .    7    .    .    .    .    .    .    .
    .    .    .    6    .    .    .    .    .    .
    .    .    .    .    .    .    .    .    .    .
    .    .    .    .    .    4    .    .    .    .
    .    .    .    .    .    .    .    .    .    .
    .    .    .    5    .    .    .    .    .    .
    .    .    .    .    .    .    .    .    1    .
    .    .    .    .    .    .    H    .    .    3
    P    .    .    .    2    .    .    .    .    .
Given that the Human intends to place an object next to the purple star at (3, 7), which object is the Human more likely to pick
up next? (a) yellow star at (4, 0). (b) red star at (3, 3).
```

*Figure 11.* An example of textual transcript for GridWorld Question Answering.

# F. Full Results of Question Answering

While Figure 4 provides an overview of the results for *MindZero* and the baselines, we present the full results for our question answering experiments across two domains in Table 4 below.

*Table 4.* Full question answering results of *MindZero* and baselines on (a) GridWorld and (b) Household domains. Best results overall and among open-weight models are shown in **bold** and underlined. * indicates methods with text-only inputs.

<table>
<tr><td colspan="3" align="center">*(a)* Gridworld Question Answering</td><td colspan="3" align="center">*(b)* Household Question Answering</td></tr>
<tr><td>Method</td><td>Accuracy ↑</td><td>TFLOPs ↓</td><td>Method</td><td>Accuracy ↑</td><td>TFLOPs ↓</td></tr>
<tr><td>Qwen3-VL-4B</td><td>37.7</td><td>**3.6**</td><td>Llama-3.1-8B</td><td>41.3</td><td>12.9</td></tr>
<tr><td>Qwen3-VL-4B-Think</td><td>42.7</td><td>67.1</td><td>Llama-3.2-3B</td><td>34.8</td><td>**4.0**</td></tr>
<tr><td>Qwen3-VL-8B</td><td>43.3</td><td>7.2</td><td>Qwen3-4B</td><td>42.8</td><td>10.9</td></tr>
<tr><td>Qwen3-VL-8B-Think</td><td>44.7</td><td>110.9</td><td>Qwen3-4B-Think</td><td>45.0</td><td>41.3</td></tr>
<tr><td>Qwen3-VL-235B-A22B</td><td>39.3</td><td>21.9</td><td>Qwen3-235B-A22B</td><td>54.5</td><td>80.4</td></tr>
<tr><td>Qwen3-VL-235B-A22B-Think</td><td>44.3</td><td>1767.5</td><td>Qwen3-235B-A22B-Think</td><td>54.0</td><td>2663.0</td></tr>
<tr><td>GPT-5.2</td><td>50.7</td><td>Proprietary</td><td>GPT-5.2</td><td>65.0</td><td>Proprietary</td></tr>
<tr><td>GPT-5.2-Think</td><td>50.7</td><td>Proprietary</td><td>GPT-5.2-Think</td><td>73.5</td><td>Proprietary</td></tr>
<tr><td>Gemini-3-Flash</td><td>68.0</td><td>Proprietary</td><td>Gemini-3-Flash</td><td>67.2</td><td>Proprietary</td></tr>
<tr><td>Gemini-3-Pro</td><td>83.7</td><td>Proprietary</td><td>Gemini-3-Pro</td><td>60.8</td><td>Proprietary</td></tr>
<tr><td colspan="3">*ThoughtTracing* (*Kim et al., 2025*)</td><td colspan="3">*ThoughtTracing* (*Kim et al., 2025*)</td></tr>
<tr><td>w/ Qwen3-VL-4B</td><td>50.3</td><td>31.0</td><td>w/ Llama-3.1-8B</td><td>44.3</td><td>571.7</td></tr>
<tr><td>w/ Qwen3-VL-8B</td><td>56.7</td><td>54.3</td><td>w/ Llama-3.2-3B</td><td>43.5</td><td>232.9</td></tr>
<tr><td>w/ Qwen3-VL-235B-A22B</td><td>53.0</td><td>169.8</td><td>w/ Qwen3-4B</td><td>54.5</td><td>291.2</td></tr>
<tr><td>w/ GPT-5.2</td><td>57.3</td><td>Proprietary</td><td>w/ Qwen3-235B-A22B</td><td>59.8</td><td>2097.9</td></tr>
<tr><td>w/ Gemini-3-Flash</td><td>64.0</td><td>Proprietary</td><td>w/ GPT-5.2</td><td>68.0</td><td>Proprietary</td></tr>
<tr><td colspan="3">*AutoToM* (*Zhang et al., 2025*)</td><td>w/ Gemini-3-Flash</td><td>72.3</td><td>Proprietary</td></tr>
<tr><td>w/ Qwen3-VL-4B</td><td>49.3</td><td>344.4</td><td colspan="3">*AutoToM* (*Zhang et al., 2025*)</td></tr>
<tr><td>w/ Qwen3-VL-8B</td><td>52.3</td><td>741.2</td><td>w/ Llama-3.1-8B</td><td>54.0</td><td>136.3</td></tr>
<tr><td>w/ Qwen3-VL-235B-A22B</td><td>44.7</td><td>1089.7</td><td>w/ Llama-3.2-3B</td><td>51.0</td><td>23.4</td></tr>
<tr><td>w/ GPT-5.2</td><td>57.3</td><td>Proprietary</td><td>w/ Qwen3-4B</td><td>54.7</td><td>177.5</td></tr>
<tr><td>w/ Gemini-3-Flash</td><td>47.0</td><td>Proprietary</td><td>w/ Qwen3-235B-A22B</td><td>67.5</td><td>389.9</td></tr>
<tr><td colspan="3">***MindZero*** (*Ours*)</td><td>w/ GPT-5.2</td><td>76.5</td><td>Proprietary</td></tr>
<tr><td>w/ Qwen3-VL-4B</td><td>**95.0**</td><td>**3.6**</td><td>w/ Gemini-3-Flash</td><td>**80.2**</td><td>Proprietary</td></tr>
<tr><td>w/ Qwen3-VL-8B</td><td>92.3</td><td>7.2</td><td colspan="3">***MindZero*** (*Ours*)</td></tr>
<tr><td></td><td></td><td></td><td>w/ Llama-3.1-8B</td><td>76.2</td><td>12.9</td></tr>
<tr><td></td><td></td><td></td><td>w/ Llama-3.2-3B</td><td>77.8</td><td>4.4</td></tr>
<tr><td></td><td></td><td></td><td>w/ Qwen3-4B</td><td>72.7</td><td>13.1</td></tr>
</table>

