# OpenReview forum: "MindZero: Learning Online Mental Reasoning With Zero Annotations"
_ICML.cc/2026/Conference — ICML 2026 regular_

### Official Review · Reviewer_HnCG · 2026-03-11

**Soundness:** 3
**Presentation:** 3
**Significance:** 3
**Originality:** 2
**Overall Recommendation:** 4
**Confidence:** 3

**Summary:**

The paper considers the problem of proactive assistance by AI agents for human users in the real world by understanding their minds and anticipating their needs. In particular, they approach this problem from a Theory of Mind perspective, where they train the agent to continuously infer and hypothesize the human user’s mental state based on the user’s behaviour. Moreover, they aim to train the agent such that it keeps track of multiple competing hypotheses about the user’s mental state, and uncertainties over all of them are constantly updated over time. However, training such models for predicting mental states requires collecting costly large-scale training data with reliable annotations of ground truth mental states. Hence, recent works leverage LLMs to propose and evaluate mental state hypotheses by combining them with standard theory of mind methods. The paper claims that such prior works are still computationally prohibitive for real-world assistance tasks and motivates their new theory of mind reasoning framework that trains LLMs to reason about mental states efficiently without using mental state annotations. Their core training idea relies on requiring models to generate mental state hypotheses and rewarding them when the generated hypotheses assign high likelihood to the ground truth actions performed. They term it Self-Supervised RL, as it eliminates the need for ground truth mental state labels and learns directly from behaviour. Empirical results show a systematic evaluation of the proposed method MindZero and other recent ToM methods on a suite of challenging online mental reasoning and proactive AI assistance benchmarks.

**Compliance With Llm Reviewing Policy:**

Affirmed.

**Final Justification:**

The authors addressed my concerns. I keep my positive rating.

**Key Questions For Authors:**

Can the authors clarify the training data construction in lines 363–374?
Can the paper more explicitly distinguish MindZero from Thought Tracing and AutoToM?
Can the authors provide more implementation details of the GRPO training? (It is also missing in appendix)

**Limitations:**

Yes

**Strengths And Weaknesses:**

Strengths:
- The paper considers an important problem of training AI agents for real-world proactive assistance in a computationally efficient manner with limited data.
- The paper sidesteps costly data annotation of ground truth mental states by designing a self-supervised RL framework that rewards generated hypotheses that correctly predict the ground truth action.
- The method also calculates posterior probabilities of the generated hypotheses to track their uncertainty based on evolving state and action, and uses this in the training reward.
- Broad evaluation against state-of-the-art ToM methods on both question answering and proactive assistance tasks shows significant improvement over prior methods.
- Small language models trained with MindZero outperform baselines while retaining similar inference cost to pretrained models, and lower inference cost compared to other methods in terms of TFLOPs.
- Ablations have been performed with respect to various modules of the proposed methodology.


Weaknesses
- Primarily, it is unclear which aspects of the proposed methodology are novel compared to prior ToM methods. Even though the paper has a concrete related work section citing relevant works, there are no clear details about how those works’ methodologies differ from this work. For example, the difference from the main experimental baselines, Thought Tracing and AutoToM, is not explained sharply enough at the algorithmic level.
- Training data construction is unclear, especially around lines 363–374 for household QA. For instance, what exactly is the candidate set? And is hypothesis filtering used during data construction or during the GRPO rollouts? Further details and examples illustrating this would be helpful (this could be moved to the appendix if space is limited; currently the appendix is very brief).

---

> ### Author Rebuttal · Authors · 2026-03-31
>
> Thank you for your insightful review and for recognizing the importance of the problem, the paper’s efficient self-supervised framework, its broad and strong empirical results, and its careful ablation study. We are happy to address your questions and suggestions below:
>
> ---
>
> > **W1 & Q2: Need Clarification on Novelty Compared with Prior ToM Methods**
>
> **ThoughtTracing** is an inference-time method that traces the mental states of specific agents by generating hypotheses and weighting them based on observations. **AutoToM** is also an inference-time method that builds an initial agent model and iteratively refines it based on uncertainty. Both require multiple calls of LLMs to complete explicit Bayesian inverse planning. In contrast, **MindZero** trains a multimodal LLM to directly produce results in a single pass.
>
> For example, given a question, **AutoToM** first extracts observable variables and builds an initial agent model. It then applies Bayesian inverse planning, computing local conditionals and estimating the posterior of the target variable by marginalizing over latent variables. It iteratively adjusts the model until utility is sufficient, then outputs the answer. In contrast, **MindZero** simply prompts the model with the question and directly produces the answer.
>
> In summary, the key differences between *MindZero* and test-time methods are:
>
> 1. **Single forward pass:** *MindZero* only needs one forward pass, making it efficient and suitable for real-time assistance, instead of relying on multi-step test-time frameworks.
>
> 2. **Training-based approach:** *MindZero* learns mental reasoning from human behavior data, improving the model’s inherent ability rather than relying on test-time search. It allows a finetuned small LLM to achieve strong inference, surpassing much larger pretrained LLMs.
>
> ---
>
> > **W2 & Q1: Need Clarification on Household QA Data Construction**
>
> We clarify the training data construction below:
>
> Hypothesis filtering is applied only during data construction for the Household QA setting. Specifically, we unroll each MMToM-QA training episode into trajectory prefixes and, for each prefix, derive four sets: seen, unseen, visited, and unvisited. Here, seen/unseen denote objects the agent has or has not observed, while visited/unvisited denote locations it has or has not checked. We then instantiate binary QA examples from these sets so the training data matches the binary-choice format used at evaluation. This yields up to four QA examples per valid prefix, using two belief templates and two goal templates.
>
> For **belief questions**, we condition on a candidate goal and ask which belief is more likely. We sample an unseen object $g$ as the hypothesized target and choose a queried container $c$ in two ways. In the first template, $c$ is the agent’s most recently visited container, so the correct answer is that the agent believes $g$ is inside $c$. In the second, $c$ is sampled from unvisited containers, so the correct answer is that the agent believes $g$ is not inside $c$.
>
> For **goal questions**, we ask which goal is more likely using two templates. In the first, we condition on a belief: we sample an unseen object $g_{\text{unseen}}$​, another object $g_{\text{alt}}$​, and use the last visited container $c_{\text{last}}$ to form a belief (e.g., “Mary does not think $g_{\text{alt}}$​ is in the fridge”). The model then chooses whether Mary aims to get $g_{\text{unseen}}$​ or $g_{\text{alt}}$​, with the correct answer being the unseen object. In the second template, we ask without an explicit belief: we sample one unseen and one seen object and ask which the agent is trying to get, again with the unseen object as correct.
>
> We will include the details in the revision.
>
> ---
>
> > **Q3: Insufficient Details for GRPO Training**
>
> In Household Question Answering, we train *MindZero* with standard GRPO in the verl framework on 4×H100 GPUs. The reward model is a separately served Qwen-235B-FP8 model running with vLLM on another 4×H100 machine. We use 16 rollout samples per prompt as the hypothesis proposal set, a rollout batch size of 16, a global batch size of 8, and train for 20 epochs with AdamW in bf16. The main optimization hyperparameters are a learning rate of $1\times10^{-6}$, weight decay of $1\times10^{-2}$, a max grad norm of 1.0, and a KL coefficient of $1\times10^{-2}$.
>
> In GridWorld Question Answering, we use the same GRPO setup with adjustments: we increase the rollout batch size to 32 and the global batch size to 32, keep 16 hypotheses per prompt, and train for 20 epochs with the same optimizer and KL settings.
> In Household Proactive Assistance and GridWorld Proactive Assistance, we use the same GRPO setup, increase both rollout and global batch sizes to 32, and reduce the number of training epochs (e.g., to 8).
>
> We will include the GRPO training details in the revision.

---

> > ### Author Rebuttal · Reviewer_HnCG · 2026-04-03
> >
> > I thank the authors for the rebuttal and the clarifications. I just have one question regarding the baselines. If your method is primarily training based, why are both the main baselines inference-time based? Are there no prior works that propose a training based method to solve this problem and could have been compared to?

---

> > > ### Author Response · Authors · 2026-04-04
> > >
> > > Thank you for your response. Our goal is to learn mental reasoning without ground-truth annotations. To our knowledge, *MindZero* is **the first learning-based ToM method that does not require ground-truth annotations**. In contrast, all prior works on learning-based ToM methods (e.g., [1,2,3,4,5]) require ground-truth annotations. A **fair comparison** would require evaluating methods **under the same setting**, so we can only compare it against non-learning-based methods that do not require annotations. This new learning ability for ToM introduced by *MindZero* eliminates the need of costly and often unreliable mental state annotation collection.
> > >
> > > [1] Rabinowitz et al., Machine Theory of Mind. ICML 2018.
> > >
> > > [2] Bortoletto et al., Neural Reasoning About Agents' Goals, Preferences, and Actions. AAAI 2024.
> > >
> > > [3] Zhang et al., Overcoming Multi-step Complexity in Multimodal Theory-of-Mind Reasoning: A Scalable Bayesian Planner. ICML 2025.
> > >
> > > [4] Sclar et al., Explore Theory of Mind: Program-guided Adversarial Data Generation for Theory of Mind Reasoning. ICLR 2025.
> > >
> > > [5] Lu et al., Do Theory of Mind Benchmarks Need Explicit Human-like Reasoning in Language Models?. arXiv preprint, 2025.

---

### Official Review · Reviewer_XJT5 · 2026-03-13

**Soundness:** 3
**Presentation:** 3
**Significance:** 3
**Originality:** 3
**Overall Recommendation:** 5
**Confidence:** 4

**Summary:**

The paper introduces MindZero, a framework for learning online Theory of Mind reasoning without any need for explicit annotations of mental states. The paper aims to address three major problems: enabling agents to maintain multiple hypotheses simultaneously, making this reasoning faster and more efficient, and addressing the lack of annotated data in this space. The paper's core hypothesis is that good mental states are those that predict observed behavior, which means models can learn online mental state inference using action-likelihood signals. MindZero learns mental states implicitly by generating candidate hypotheses first, which are then scored using a planner. Basically, the planner predicts actions that would follow from these hypotheses, which are then compared against observed actions. Training is performed using an RL objective that rewards hypotheses that explain observed behaviors. Experiments are done in a gridworld setting and a household assistance environment evaluating on two tasks: a Q&A on human mental state and proactive assistance to the human agent. The empirical results indicate strong performance over pretrained models and baselines.

**Compliance With Llm Reviewing Policy:**

Affirmed.

**Final Justification:**

The authors acknowledged some concerns I raised about their claims regarding RL, and, in light of the additional empirical evidence presented in the rebuttal, I have updated my score from weak accept to accept.

**Key Questions For Authors:**

1. The method is described as reinforcement learning, yet the objective resembles variational inference over latent variables with an externally computed reward; could the authors clarify whether RL is conceptually necessary for the framework or primarily serves as a practical optimization mechanism for the non-differentiable objective?

2. Since the reward signal depends on environment-specific likelihood estimates, does that mean that under the current formulation, the generalization is conditional on the likelihood model?

**Limitations:**

The authors can further clarify that the system learns behaviorally consistent explanations rather than recovering the true underlying mental states. Also, the authors could clarify that the framework focuses on goal inference from behavior, which is only a component of Theory-of-Mind.

**Strengths And Weaknesses:**

# Strengths:
1. The problem is well motivated. Many existing works rely on symbolic methods, which are slow or supervised learning with annotated data, where such annotations are rarely available in reality. Addressing this gap is meaningful for interactive assistants and embodied AI.

2. The proposed reward signal is intuitive. Hypotheses are rewarded when they assign a high likelihood to observed behavior. Linking mental state inference to maximizing action likelihood under a planner gives a nice bridge between inverse planning and language model training.

3. Empirical results are strong in both the QA task and the proactive assistance task. In the household domain, smaller models trained with MindZero outperform much larger proprietary models.

4. Ablations show that maintaining uncertainty instead of committing to a single inferred goal benefits the model.

# Weaknesses:
1. Although the method is framed as reinforcement learning. RL is primarily used as an optimization mechanism because the model outputs discrete textual hypotheses. However, the paper sometimes presents RL as the core conceptual innovation rather than an implementation choice.

2. Some of the claims are largely empirical. For example, the claim that the model internalizes Bayesian inverse planning, but there is no theoretical analysis of how closely the learned distribution approximates the true posterior.

3. The reward is heavily and critically tied to the likelihood model implemented using planners or LLMs. It is very likely that this likelihood model maybe biased or permissive which can encourage spurious explanations for the observed behavior. This specifcally critical in embodied settings like household where LLMs are used to estimate actions likelihoods.
4. Given that uncertainty modeling is a key claimed advantage of the framework, calibration and uncertainty quality should be examined more directly. This seems to be missing.

---

> ### Author Rebuttal · Authors · 2026-03-31
>
> Thank you for your insightful review and for recognizing the strong motivation of the problem, the intuitive and principled reward design, the strong empirical results, and the useful ablation findings. We will address your questions and suggestions below:
>
> ---
> >**W1 & Q1: Clarifying the Role of RL**
>
> We would like to clarify the role of RL in *MindZero*.
>
> (1) **RL primarily serves as a practical training mechanism.** During training, the model generates textual hypotheses about latent mental states and receives rewards when these hypotheses better explain the actions humans actually take. In this sense, we follow the common convention in LLM training of treating reward-based optimization methods such as GRPO as RL.
>
> (2) **Our main contribution is the self-supervised reward design in RL, not RL itself.** *MindZero* learns from behavioral signals rather than ground-truth mental state labels, enabling it to learn directly from human behavior.
>
> (3) **RL optimizes the inference process.** Our ultimate goal is mental-state inference. To this end, we use RL as an optimization method to improve the model’s inference ability through reward feedback.
>
> ---
> >**W2: Largely Empirical Claims Without Theoretical Analysis**
>
> Our method is theoretically connected to Bayesian inverse planning: our reward is derived from the ELBO (Equation (3)), so optimizing the SSRL objective (Equation (4)) corresponds to optimizing the same variational objective used to approximate the Bayesian posterior under our formulation.
>
> ---
> >**W3 & Q2: Sensitivity to Bias in the Likelihood Model**
>
> We use different planners or LLMs across domains to mitigate this concern, as prior works have shown that model-based methods with planners and LLM backends have strong performance in GridWorld [1] and Household [2].
>
> Your understanding of generalization is insightful. *MindZero* aims to internalize the inference capabilities of model-based inverse planning when that signal is reliable. As a result, generalization depends on the quality of the likelihood model.
>
> In practice, for a new domain, one can first verify whether the underlying model-based method provides reliable likelihood estimates. If not, a practical solution is to collect a small amount of human data (e.g., examples or guidance) to improve the likelihood signals.
>
> [1] Baker et al., Action Understanding as Inverse Planning. Cognition, 2009.
>
> [2] Zhang et al., AutoToM: Scaling Model-based Mental Inference via Automated Agent Modeling. NeurIPS, 2025.
>
> ---
> >**W4: Lack of Direct Analysis of Calibration and Uncertainty**
>
> In proactive assistance experiments, the assistant must perform online goal inference with uncertainty estimation, identifying the user’s goal early enough to provide timely help, but not so early that it commits to an incorrect hypothesis that may incur large penalties. The strong results highlight *MindZero*'s advantage in uncertainty estimation.
>
> To further assess uncertainty estimation, we examine the predicted probabilities of the hypotheses, which should ideally match those from explicit Bayesian inverse planning (BIP). We report the KL divergence between *MindZero*-predicted and BIP probabilities for Gridworld assistance below. This KL divergence is significantly lower than that of the pretrained model, indicating improved uncertainty estimation.
>
> | Progress | KL (*MindZero* w/ Qwen3-VL-4B ‖ BIP) | KL (Qwen3-VL-4B ‖ BIP)
> | :-- | ---: | ---: |
> | 25% | 10.67 | 17.89 |
> | 50% | 6.48 | 13.96 |
> | 75% | 3.33 | 5.50 |
> ---
> >**L1: Clarify the Scope of Mental-State Inference**
>
> In online goal inference, many mental state hypotheses can explain the observed behavior, so we fully understand your concern about learning behaviorally consistent explanations rather than true mental states. To address this:
>
> (1) We conduct **explicit evaluations via question answering** in Sections 5.1 and 5.3, showing that *MindZero* achieves SOTA accuracy in directly answering questions about mental states.
>
> (2) We conduct additional analysis and will add a subsection, “5.6 Online Goal Inference Dynamics,” and “Figure 5: Accuracy for online goal inference versus task progress” (see [our anonymous link](https://anonymous.4open.science/r/figure-BFB5/Accuracy_for_online_goal_inference.png)). In both assistance settings, *MindZero* **steadily improves its goal prediction over time**. In GridWorld, it is the only method whose accuracy rises substantially as the task unfolds, eventually reaching a strong level, while baselines **remain very low** until late in the trajectory. In Household, *MindZero* achieves the best performance, with consistently improving accuracy, significantly outperforming base models and matching much larger pretrained models.
>
> Regarding “focusing on goal inference,” we also evaluate **belief inference** in Section 5.3. Additionally, as shown in AutoToM, model-based inference applies to goals, beliefs, and preferences, which can likewise be incorporated into the *MindZero* framework.

---

> > ### Author Rebuttal · Reviewer_XJT5 · 2026-04-03
> >
> > The rebuttal clarifies that RL is primarily an optimization mechanism and strengthens the empirical evaluation of uncertainty. However, key theoretical questions remain unresolved, particularly regarding the precise relationship between the training objective and variational inference, as well as the dependence of the learned inference procedure on the likelihood estimator. As such, while the paper presents a promising and empirically strong approach, its theoretical claims remain somewhat overstated. Overall, the rebuttal improves the paper's clarity and strengthens certain empirical aspects. I increase my score.

---

### Official Review · Reviewer_5XEX · 2026-03-24

**Soundness:** 2
**Presentation:** 2
**Significance:** 3
**Originality:** 3
**Overall Recommendation:** 4
**Confidence:** 3

**Summary:**

The paper studies the problem of inferring human mental states from their behavior. To this end, the authors propose a self-supervised
RL method, MindZero, that trains the model without mental state annotations. The proposed MindZero learns directly from behavioral data in a self-supervised manner. For experiments, the authors have conducted experiments and evaluations on four different settings, including GridWorld Question Answering, GridWorld Proactive Assistance, Household Question Answering and Household Proactive Assistance. Experiments show the proposed method achieves a better performance than previous methods, while also achieving a better TFLOPs. The evaluation of MindZero and other methods under these settings is claimed as another major contribution of the paper.

**Compliance With Llm Reviewing Policy:**

Affirmed.

**Final Justification:**

The rebuttal has addressed most of my concerns. Therefore, I am inclined to accept the paper.

**Key Questions For Authors:**

I would suggest the authors to address the weaknesses in the rebuttal, especially:

1. For weakness 1, what is the efficiency measured by latency going to be like for the tables?

2. For weakness 2, what is the performance of Gemini models in Table 1?

**Limitations:**

Yes

**Strengths And Weaknesses:**

Strengths:

1. The paper studies an important problem - predict human mental states from their behavior. I think the problem should have lots of downstream applications.

2. I appreciate the approach of the paper. It makes sense to me to develop such a self-supervised method to address the challenge of limited ground-truth mental state annotations in real-world domains.

Weaknesses:

1. Evaluation of efficiency. In the experiments, the metric reported to measure efficiency is TFLOPs. It is better to also report latency, as this is an important metric to reflect the efficiency at the actual inference time.

2. Evaluation of Gemini on Table 1. As the authors claim the extensive evaluation is one of their major contributions, it would be better to also include the evaluation of Gemini models as in Table 3.

3. Lack of qualitative results or demos. In Figure 3, the authors present an illustration of four different evaluation settings; however, there is no qualitative results of the proposed method compared with others in this paper. It would be better to also include this to give the audience a better and more intuitive understanding of what the performance improvement looks like.

4. Typos. In Line 290, Line 311, Line 380, pretrained should be 'pre-trained'.

---

> ### Author Rebuttal · Authors · 2026-03-31
>
> Thank you for your insightful review and for recognizing the importance of the problem and the value of our self-supervised approach for learning mental state inference in real-world settings with limited annotations. We will address your questions and suggestions below:
>
> ---
>
> > **W1 & Q1: Better to Report Latency for Inference Efficiency**
>
> We did not report latency since different models require different numbers of GPUs, making a fair comparison difficult.
>
> Having said that, we have added latency evaluation and will include the tables below in the revision to report latency. We find that latency is generally highly correlated with TFLOPs (model size × number of generated tokens). *MindZero* trained with Qwen models also consistently have the **lowest latency**. Note that we focus on latency comparison among single-pass methods. Model-based methods (ThoughtTracing and AutoToM) require multiple calls of LLMs, which will have much higher latency.
>
> Average latency for gridworld assistance:
>
> | Method                     | Latency (s) |
> |:--------------------------|------------:|
> | Qwen3-VL-4B              | 15.49       |
> | Qwen3-VL-8B              | 13.32       |
> | Qwen3-VL-235B            | 34.71       |
> | Gemini-3-Flash           | 158.32      |
> | *MindZero* w/ Qwen3-VL-4B  | 20.02       |
> | *MindZero* w/ Qwen3-VL-8B  | **11.87**       |
>
> Average latency for household assistance:
>
> | Method                         | Latency (s) |
> |:------------------------------|------------:|
> | Llama-3.1-8B                  | 110.75      |
> | Llama-3.2-3B                  | 82.88       |
> | Qwen3-4B                      | 48.98       |
> | Qwen3-235B                    | 146.97      |
> | GPT-5.2                       | 120.66      |
> | Gemini-3-Flash                | 63.06       |
> | *MindZero* w/ Llama-3.1-8B      | 111.60      |
> | *MindZero* w/ Llama-3.2-3B      | 85.88       |
> | *MindZero* w/ Qwen3-4B     | **33.90**   |
>
> ---
>
> > **W2 & Q2: Better to Include Gemini Evaluation in Table 1**
>
> Thanks for pointing this out. We have added the experiments on Gemini. We show the updated Table 3 below and will update the corresponding part in Section 5.3.
>
> | Method | Accuracy ↑ | TFLOPs ↓ |
> |---|---:|---:|
> | Llama-3.1-8B | 41.3 | 12.9 |
> | Llama-3.2-3B | 34.8 | 4.0 |
> | Qwen3-4B | 42.8 | 10.9 |
> | Qwen3-4B-Think | 45.0 | 41.3 |
> | Qwen3-235B-A22B | 54.5 | 80.4 |
> | Qwen3-235B-A22B-Think | 54.0 | 2663.0 |
> | GPT-5.2 | 65.0 | Proprietary |
> | GPT-5.2-Think | 73.5 | Proprietary |
> | Gemini-3-Flash | 67.2 | Proprietary |
> | Gemini-3-Pro | 60.8 | Proprietary |
> | **ThoughtTracing (Kim et al., 2025)** |  |  |
> | + Llama-3.1-8B | 44.3 | 571.7 |
> | + Llama-3.2-3B | 43.5 | 232.9 |
> | + Qwen3-4B | 54.5 | 291.2 |
> | + Qwen3-235B-A22B | 59.8 | 2097.9 |
> | + GPT-5.2 | 68.0 | Proprietary |
> | + Gemini-3-Flash | 72.3 | Proprietary |
> | **AutoToM (Zhang et al., 2025)** |  |  |
> | + Llama-3.1-8B | 54.0 | 136.3 |
> | + Llama-3.2-3B | 51.0 | 23.4 |
> | + Qwen3-4B | 54.7 | 177.5 |
> | + Qwen3-235B-A22B | 67.5 | 389.9 |
> | + GPT-5.2 | 76.5 | Proprietary |
> | + Gemini-3-Flash | 80.2 | Proprietary |
> | **MindZero (Ours)** |  |  |
> | + Llama-3.1-8B | 73.7 | 12.9 |
> | + Llama-3.2-3B | 76.0 | 4.4 |
> | + Qwen3-4B | 73.8 | 13.1 |
>
> ---
>
> > **W3: Lack of Qualitative Comparisons or Demos**
>
> Thank you for your suggestions.
>
> We will add the following figures on qualitative results in the revised experiment results section:
>
> (1) In (a) GridWorld and (b) Household Proactive Assistance, we will show how *MindZero* improves compared to the pre-trained model using a **side-by-side comparison**. As the helper observes the human’s actions over time, we will present the probability distribution over multiple goal hypotheses maintained by *MindZero* and the pre-trained model, where *MindZero* better performs online goal inference under ambiguity, identifying the user’s goal **early enough to provide timely assistance**.
>
> (2) We will illustrate how training works, using concrete examples of mental state hypotheses, their probabilities, action likelihoods, and how we compute the reward from these.
>
> ---
>
> > **W4: Typos**
>
> Thanks for pointing this out. We will fix it.

---

> > ### Author Rebuttal · Reviewer_5XEX · 2026-04-01
> >
> > Thank the authors for the rebuttal. The rebuttal has addressed most of my concerns. I am therefore leaning to accept the paper.

---

> > > ### Author Response · Authors · 2026-04-05
> > >
> > > Thank you for your insightful review and for your recognition of our paper. We are very glad that we have fully addressed your concerns, and would greatly appreciate it if you could consider raising the score.

---

### Decision · Program_Chairs · 2026-04-30

**Decision:**

Accept (regular)

**Comment:**

This paper tackles an important, practical problem in proactive AI assistance and embodied agents. It eliminates need for annotated mental states via an intuitive action-likelihood reward; efficiently maintains and updates multiple hypotheses with uncertainty and provides strong empirical results across tasks, outperforming baselines and larger models.

In particular, the paper introduces MindZero, a framework for training AI agents to perform online Theory of Mind (ToM) reasoning without requiring annotated mental state data. Instead of relying on costly supervision, MindZero uses a self-supervised RL objective where the model generates multiple candidate mental state hypotheses and rewards those that best predict observed human actions. A planner evaluates each hypothesis by estimating the likelihood of resulting actions, enabling the model to maintain and update uncertainty over multiple competing hypotheses efficiently.

The approach addresses key challenges in ToM reasoning: avoiding expensive annotations, improving computational efficiency, and enabling parallel hypothesis tracking. Experiments in gridworld and household assistance settings show that MindZero achieves strong performance on both mental state QA and proactive assistance tasks, outperforming prior ToM methods and even larger pretrained models while maintaining lower inference cost.

During the rebuttal, there are a few minor issues and the authors have mostly addressed them.

Therefore we recommend accepting the paper.